# The Fusion of MRI and CT in the Planning of Brachytherapy for Cancer of the Uterine Cervix

**Roland Merten** [1,*], **Mirko Fischer** [1], **Hans Christiansen** [1], **Kristina I. Ringe** [2], **Rüdiger Klapdor** [3] and **Jörn Wichmann** [1]

1   Clinic for Radiation Therapy and Special Oncology, Hannover Medical School, 30625 Hannover, Germany; Fischer.Mirko.Ambulanzzentrum@mh-hannover.de (M.F.); Christiansen.Hans@mh-hannover.de (H.C.); Wichmann.Joern@mh-hannover.de (J.W.)
2   Institute for Radiology, University Hospital Hannover, 30625 Hannover, Germany; ringe.kristina@mh-hannover.de
3   Department of Gynecology and Obstetrics, Hannover Medical School, 30625 Hannover, Germany; Klapdor.ruediger@mh-hannover.de
*   Correspondence: merten.roland@mh-hannover.de

**Abstract:** Introduction: tumors of the uterine cervix are among the most common carcinomas in women. Intracervical brachytherapy is an indispensable part of curative treatment. Although the tumor is significantly more recognizable in MRI than in CT, the practical application of MRI in brachytherapy planning is still difficult. The present study examines the technical possibilities of merging CT and MRI. Materials and Methods: the treatment files and imaging of all 53 patients who had been irradiated by image-guided adaptive brachytherapy (IGABT) between January 2019 and August 2021 at the Department of Radiotherapy of the Hannover Medical School were evaluated, retrospectively. Patients were treated first with an external beam radiotherapy (EBRT) combined with simultaneous chemotherapy. After an average of 4.2 weeks, the preparation for IGABT began. The clinical target volume (CTV) for brachytherapy was contoured first in an MRI acquired before starting EBRT (MRI 1) and once more in a second MRI just before starting IGABT (MRI 2). Then, after inserting the intravaginal applicator, a CT-scan was acquired, and the CTV was contoured in the CT. Finally, the recordings of MRI 1, MRI 2, and the CT were merged, and the congruence of CTVs was quantitatively evaluated. Results: the CTV delineated in MRI 2 was, on average, 28% smaller than that in MRI 1 after an average applied radiation dose of 42 Gy. The CTV delineated in the CT covered an average of no more than 80.8% of the CTV delineated in MRI 2. The congruence of CTVs was not superior in patients with a smit sleeve in the cervical channel, with a 3D-volumetric MRI or with a contrast-enhanced sequence for MRI. Conclusion: the anatomical shape and position of the uterus is significantly changed by introducing a vaginal applicator. Despite the superior delimitability of the tumor in MRI, brachytherapy cannot be reliably planned by the image fusion of an MRI without a vaginal applicator.

**Keywords:** cervical cancer; radiotherapy; brachytherapy; MRI; IGABT

## 1. Introduction

Tumors of the uterine cervix are among the most common carcinomas in women and cause fatalities in many cases [1,2]. While excision is performed for tumors in early stages, intracervical brachytherapy is an indispensable part of curative treatment in locally advanced tumors [3]. Although the tumor is more recognizable in an MRI than in a CT, the practical application of MRIs in planning brachytherapy is still difficult [4–8] for hospitals not equipped for treatment planning on MRI-only. Even in compliance with the tried and tested recommendations of the Gyn-GEC-ESTRO working group for 3D image-based anatomy [9] and for useful MRI-sequences [10], discrepancies between different modalities of imaging are challenging for Radiation oncologists. Additionally, missing the target of

Brachytherapy can easily be disastrous for the patient. Technical solutions are not always reliable. The present study examines the technical possibilities and practical applications of merging CT and MRI in the planning of brachytherapy.

## 2. Materials and Methods

The treatment files and imaging of all 53 patients who had been irradiated by image-guided adaptive brachytherapy (IGABT) [11] between January 2019 and August 2021 in the Department of Radiotherapy of the Hannover Medical School were evaluated retrospectively. Before a decision about treatment, MRI 1, of the pelvis, was acquired in the patients' hometowns using a broad variety of 1.5 T or 3 T MRI sequences, depending on the local radiologists' preferences [12]. The use of a gadolinium-based contrast agent was optional for pelvic imaging. All MRI 1 images were reviewed centrally by the same board-certified in-house-radiologist. For the staging of distant metastases, all patients received a chest CT. The upper abdomen was imaged with either contrast-enhanced CT or MRI. Patients were classified according to the TNM system, 8th edition [13,14]. A biopsy for histological proof of malignancy and to check HPV status was obtained from the tumor in all cases. According to gynecological guidelines [15], surgical staging of the pelvic lymphatic nodes was performed. All cases were reviewed by a board of in-house gynecologists, a pathologist, a radiologist, a radiation oncologist, and a medical oncologist to make treatment decisions and to allocate the patients to operations, radiotherapy, or chemotherapy, depending on TNM staging [14,15]. Informed consent of each patient was obtained before treatment and for the later retrospective analysis.

Patients allocated to radiotherapy were treated first with an external beam radiotherapy (EBRT) of 50 Gy in 25 fractions, 5 fractions per week, combined with a simultaneous chemotherapy consisting of Cisplatin 40 mg/m$^2$ weekly [16]. EBRT was 3D-planned on a pre-treatment CT-Scan using Monaco planning software (Elekta, Stockholm, Sweden) with Monte Carlo dose planning algorithm and was applied by a linear accelerator (Versa HD, Elekta, Stockholm, Sweden) with image guidance by cone beam CT (IGRT) and the volume modulated arc technique (VMAT). In the course of the EBRT series, preparation for IGABT [17] began by implanting a Smit sleeve into the cervical channel [18]. The clinical target volume (CTV) [19] for brachytherapy was contoured first in an MRI acquired before starting EBRT (MRI 1) and once more in a second MRI just before starting IGABT (MRI 2). Gross target volume (GTV) in MRI was contoured enclosing all macroscopic residuals of the tumor known from clinical examination and all suspicious T2-hyperintense areas within the uterus. CTV was contoured enclosing GTV and all residual T2-grey zones in parametria, uterine corpus or vagina, and the entire cervix. All contouring was performed in compliance with the recommendations of the Gyn-GEC-ESTRO Working group [20].Then, after inserting the intravaginal applicator (Vienna Tandem-Ring-Applicator or Geneva Tandem-Ovoid-Applicator, Elekta, Stockholm, Sweden), a helical CT-scan (Siemens Somatom 16 slices) was acquired, and the CTV was contoured in the CT. Contouring in CT was performed in knowledge of all MRI and clinical information, again in compliance with the recommendations of the Gyn-GEC-ESTRO working group [20]. Finally, the recordings of MRI 1, MRI 2, and CT were merged, and the congruence of CTVs was quantitatively evaluated. Calculating was performed within an Excel file (Microsoft, Redmond, WA, USA), which is available from the link below (Supplementary Materials). Merging was performed using Monaco software for treatment planning first (Elekta) and careful correction by hand until best possible concordance was obtained. To minimize inter-observer variability, all CTVs were contoured by the same board-certified radiologist [21] by mutual agreement with the executing brachytherapist. IGABT started an average of 4.2 weeks after the beginning of EBRT. MRI 2 was acquired in different sequences according to the choice of the local radiologists. Gadolinium contrast was optional in MRI 2.

## 3. Results

Between January 2019 and August 2021, 53 patients were treated with intracervical IGABT. Six patients received brachytherapy without adaptive MRI planning and were excluded from quantitative evaluation. Forty-seven patients could be evaluated. The patients were between 29 and 80 years old (average 52 years). Of the patients, 18.9% had cT1 carcinoma, 45.3% had cT2 carcinoma, 28.3% had cT3 carcinoma, and 7.5% had cT4 carcinoma; 34% had no metastases in regional pelvic lymph nodes N0, while 66% had metastases in regional lymph nodes N1; and 73.6% had no distant metastases M0, while 26.4% had paraaortic lymph node metastases pM1 (Lym). Patients with visceral metastases were not among the treated. None of the patients had highly differentiated carcinoma G1, 51% of patients had G2 carcinoma, while 49% had G3 carcinoma; 90.6% of patients had no invasion of lymphatic vessels L0, while 9.4% of patients had L1; 86.8% of patients had V0, while 13.2% of patients had infiltration of blood vessels V1; and 86.8% of these cases were squamous cell carcinoma, 9.4% were adenocarcinoma, and 3.8% were small cell or adenosquamous.

For the acquisition (vibe, thrive, LAVA, and T2-space) of MRI 1, 69.8% of patients had a 3D-volumetric sequence. For MRI 2, it was only 25%; 47% of patients had a Smit sleeve implanted in MRI 2, while 53% did not, and 36% had Gadolinium contrast in MRI 2, while 64% had none. The patient characteristics are listed in Table 1.

**Table 1.** Patient characteristics.

| Patient Properties | |
|---|---|
| Number of Patients | 53 |
| Adaptive MRI available | 47 |
| Age | mean 52 Y<br>min. 29 Y<br>max. 80 Y |
| Tumor size TNM | 18.9% cT1<br>45.3% cT2<br>28.3% cT3<br>7.5% cT4 |
| Pelvic lymph node metastasis | 34% N0<br>66% N1 |
| Distant lymph node metastasis | 73.6% no<br>26.4% yes |
| Grading | 0% G1<br>51% G2<br>49% G3 |
| Lymphatic vessel invasion | 9.4% L1<br>90.6% L0 |
| Blood vessel invasion | 86.8% V0<br>13.2% V1 |
| Differentiation | 86.8% squamous<br>9.4% Adenocarcinoma<br>3.8% small cell or adeno-squamous |

The EBRT began, on average, 51 days after histological proof of diagnosis (range 11 to 145 days). The volume of the CTV in MRI 1 was an average of 48 cm$^3$ (range 17 to 223 cm$^3$); in MRI 2, it was 34 cm$^3$ (range 7 to 104 cm$^3$). The decrease in CTV thus amounted to an average of 28% (range 9% progression to 75% remission of CTV) after a previously administered dose of an average of 41.96 Gy (range 12 Gy to 50 Gy, depending on the elapsed time from beginning of EBRT to MRI 2). The CTV for the IGABT in the CT was, on average, 51 cm$^3$ (range 9 to 134 cm$^3$). The volume of the intersection between MRI 2 and

the CT was, on average, 28 cm$^3$ (range 4 to 88 cm$^3$). The intersection covered an average of only 80.8% of the CTV in MRI 2 (range 39 to 100%). In other words: if the IGABT had been performed after planning in MRI 2 only, 19.2% of the tumor would have been missed by IGABT. In the group of patients with a sleeve implanted, the intersection covered an average of 84% of the CTV in MRI 2; without the sleeve, it was 78%. The coverage in the group of patients with a 3D-volumetric MRI 2 was 79%. In patients with the Gadolinium contrast agent during MRI 2, the coverage was 78% (variance 161, standard deviation 13).

The CTV was, on average, 72% larger in the CT than it was in MRI 2. Taken the other way, an average of 53% of the CTV in the CT was covered by intersection with MRI 2 (range 19 to 87%). In patients with a sleeve, an average of 52% was covered; for without a sleeve, an average of 53% was covered. Likewise, with a 3D-volumetric MRI 2, it was 57% covered, and with a contrast medium in the MRI 2, it was 58% covered.

### 4. Discussion

Due to the average reduction in the CTV by 28% during EBRT before starting IGABT, it is not possible to plan an adaptive CTV based on the pretherapeutic MRI 1. To use an MRI for adaptive planning, a second MRI is mandatory, preferably in the fifth week of EBRT.

Although CTVs have larger outlines in the CT due to the diagnostic uncertainty, after carefully merging the images in planning software, only 80.8% of the CTV was defined precisely in MRI 2. If the IGABT had been planned exclusively according to the MRI 2-Scan, 19.2% of the CTV would not have been hit. The planning of IGABT on the basis of MRI 2, therefore, does not reach sufficient coverage of the tumor according to the image fusion, when the CT was acquired with a vaginal applicator and the MRI was acquired without.

Figures 1–3 show that the uterus is often straightened, erected, and cranially shifted by the insertion of a vaginal applicator. These anatomical changes in the configuration of the uterus clearly caused the insufficient results of the CT and MRI image fusion in our study. Planning brachytherapy exclusively on the basis of an MRI can therefore only be carried out on an MRI with the vaginal applicator system in situ to enable good irradiation of the CTV and, at the same time, protection of neighboring organs at risk. The beginning of this practice has already been described in older methodological works, which, although they used image fusion, have not yet provided for the 3D volumetric planning of radiation dose application [6,8,22]. In order to use the automatic fusion of MRI and CT [23], these anatomical changes must be taken into account. That is why a vaginal applicator in situ cannot be dispensed with in scanning MRI.

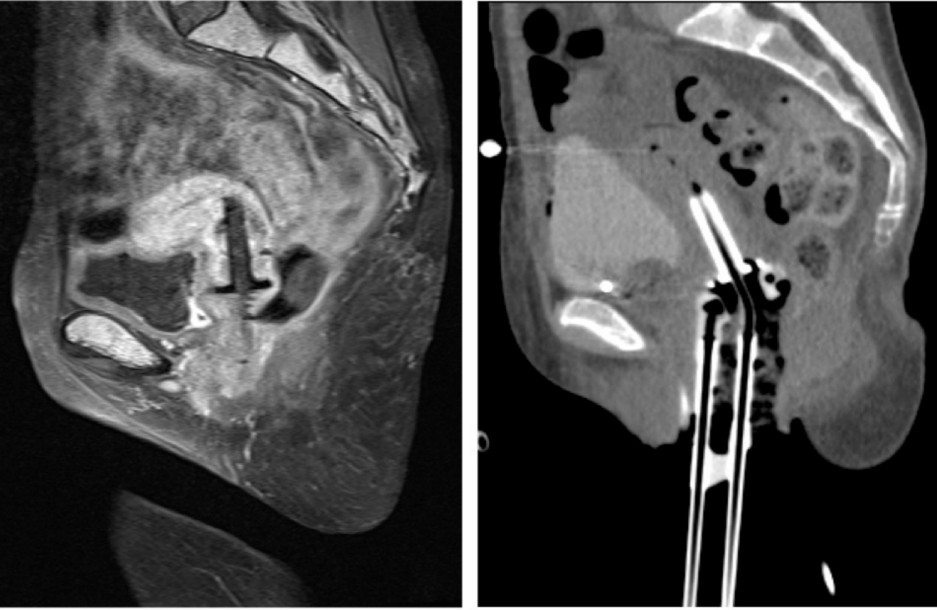

**Figure 1.** A uterus straightened by the insertion of the vaginal applicator.

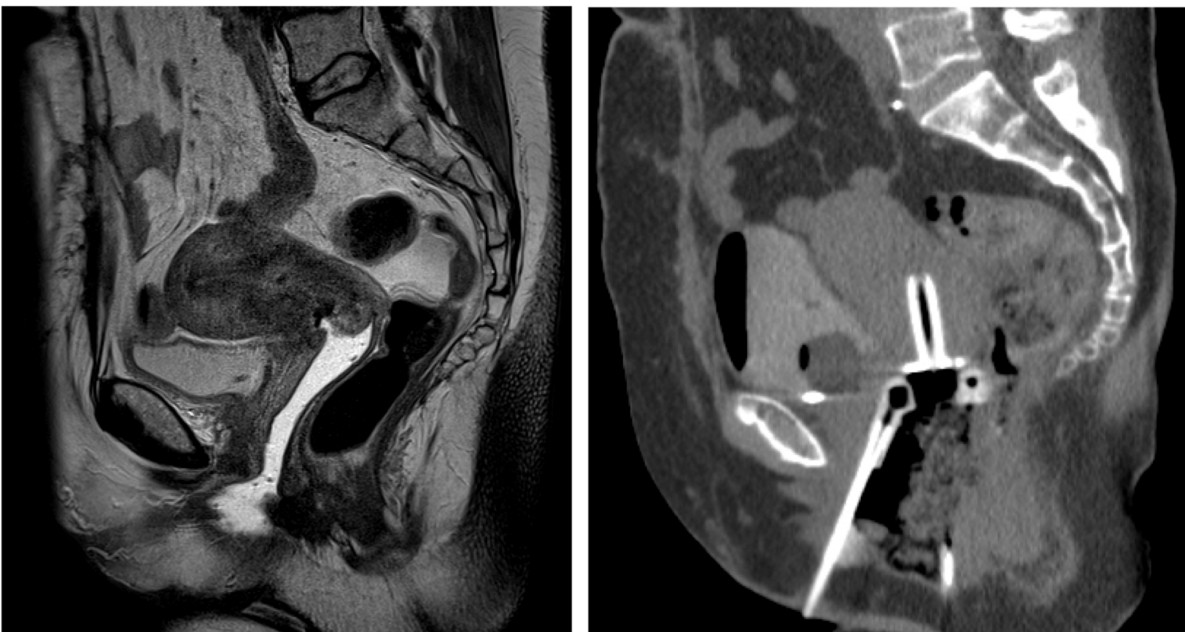

**Figure 2.** A uterus erected by the insertion of the vaginal applicator.

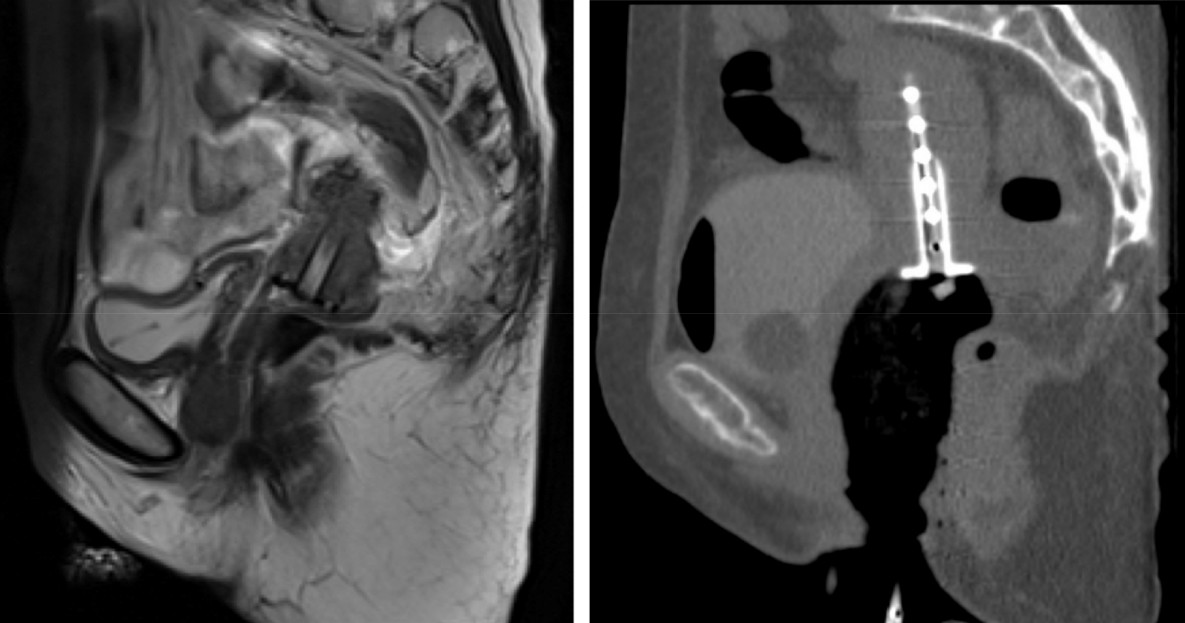

**Figure 3.** A uterus cranially shifted by the insertion of the vaginal applicator.

The superior delimitability of tumors in Gadolinium contrast-enhanced MRI described by Balcacer [24] could not be confirmed as useful for merging images by the small number of cases of our study. Although an implanted Smit sleeve improved the anatomical orientation and shape of uterus, its usage did not lead to superior results of the image fusion in our study. Neither implantation of the Smit sleeve nor use of Gadolinium contrast in MRI 2 could compensate for anatomical mismatch caused by missing vaginal applicator in MRI.

However, even without automatic fusion, the diagnostic information of MRI is indispensable [8]. Sufficient experience about suitable MRI sequences is described in other studies [12,25].

## 5. Conclusions

By introducing the vaginal applicator for brachytherapy, the anatomical shape and position of the uterus changes. Despite the diagnostically clearly better delimitability of the tumor during MRI, brachytherapy cannot be reliably planned by image fusion of CT and MRI without a vaginal applicator being inserted in both.

**Supplementary Materials:** Excel-sheet with original source data of this study is available online at https://www.mdpi.com/article/10.3390/app12020634/s1.

**Author Contributions:** Conceptualization, R.M. and H.C.; methodology, R.M., J.W., R.K., K.I.R. and M.F.; software and formal analysis, M.F.; investigation, R.M. and J.W.; visualization, K.I.R.; writing—original draft preparation, R.M.; writing—review and editing, H.C. and R.K.; funding acquisition, H.C. and R.M. All authors have read and agreed to the published version of the manuscript.

**Funding:** We acknowledge support by the German Research Foundation (DFG) and the Open Access Publication Fund of Hannover Medical School (MHH).

**Institutional Review Board Statement:** The study was conducted according to the guidelines of the Declaration of Helsinki, and approved by the Institutional Ethics Committee) of Hannover Medical School (protocol code Nr. 10031_BO_K_2021, Date 12 October 2021).

**Informed Consent Statement:** Informed consent was obtained from each single Patient before involved in the study.

**Data Availability Statement:** This study does not report publicly archived datasets. Original Excelsheet of this study is available from the corresponding author upon request.

**Conflicts of Interest:** The authors declare no conflict of interest. The funders had no role in the design of the study; in the collection, analyses, or interpretation of data; in the writing of the manuscript, or in the decision to publish the results.

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
