# Peer review of "The Fusion of MRI and CT in the Planning of Brachytherapy for Cancer of the Uterine Cervix"

_applsci, doi:10.3390/app12020634_

Round 1

Reviewer 1 Report

I like the idea behind this manuscript very much. The role of modern imaging in efficient brachytherapy is unquestionable, but there are a lot of questions to be answered. One has been approached by authors successfully. 

My concern is that introduction is relatively short; I'd call it too economical. As a clinician, I'd like to read more about the problem (as a researcher, I understand it). The results are a little cloudy, but the authors had to cope with many data in numbers. I think using a "vaginal" applicator instead of tandem with ovoids or ring applicator is a little bit confusing. I'd like to know which specific applicators were used.
IMO the manuscript is worthy of being published after suggested changes.

Author Response

Dear Madam, dear Sir,

Thank you very much for your helpful advice. First I have to apologize for forgetting to remove the names of all patients from the Excel-sheet. I corrected my mistake.

We now described more details in the introduction to clarify the clinical problem focused in our study. And we described our methods of contouring CTV in compliance with several recommendations in literature. We are using two different types of vaginal applicators, Elekta-Geneva and Elekta-Vienna, now also mentioned in the text. We do not fail to see the problem of inter-observer variability and want to avoid this problem in our study. Analysis of inter-observer variability was not the scope of this study.

Table 1 was designed to give a quick overview of patient characteristics. The many numbers in our results are unsuitable for a tabular representation, as they ask for an explanation and discussion in the text. Of course, Gadolinium improves MRI-contrast, but not even Gadolinium can compensate for the anatomical mismatch between imaging with and without vaginal applicator.

The figures show three typical variants of anatomy caused by introducing applicators. I corrected the numbers in the label. They are no longer confusing now.

best regards

Roland Merten

Reviewer 2 Report

  1. Supplementary file involved patients' informations that had not to be opened. As scientist, the authors had serious mistake.
  2. In this manuscript, the contents of Title, Abstract, Material and Methods, Results, Discussion and Conclusion are not correct each. The authors must clarify about what the authors want to show in this manuscript and subscribe along them.
  3. The introduction did not provide sufficient background.
  4. What was the definition of CTV in this manuscript? The authors compared the CTV volumes from MRI1, MRI2 and CT. However, the authors did not subscribed how to make the CTVs in this manuscript. Had the authors contoured the CTVs by the "some routine"? The authors must subscribe the definition of CTV in this manuscript and how to make the CTVs. 
  5. In this manuscript, the authors subscribed " to minimize inter-observer variability, all CTVs were contoured by the same radiologist". However, I don't think this method is good for this manuscript. It is more suitable that three or four radiologists contour all the CTVs and compare the average of them for this manuscript.
  6. Was Table 1 important for this manuscript? The authors would want to show the other results, I think, so the authors had to show other Table(s) in the Results.
  7. Why are there too much differences in the EBRT-dose (range 12Gy to 50Gy)? Because the authors compared the CTV volumes, the authors have to minimize the difference of radiation dose.
  8. Discuss the reason the coverage in the patients with Gadolinium contrast-MRI. Did Gadolinium disturb for contouring CTV?
  9. All the Figures look same. And the authors showes 2-Figure 1. Correct second Figure 1 to Figure 3.
  10. The content of Conclusion was not sufficient.

Author Response

(The authors gave the same response as above.)

Reviewer 3 Report

The paper is well written; authors should pay attention to the following:

  • The introduction is very short and requires further explanation and a review of the other researches.
  • What is the use of Elekta in Sweden? Please explain more. Is only Monaco software used?

Author Response

(The authors gave the same response as above.)

Round 2

Reviewer 2 Report

The authors revised this manuscript as the reviewer had pointed.

Author Response

Dear Madam, dear Sir, in chapter "materials and methods" I added further details, how we managed the delineation of CTV together. In chapters "results" and "discussion" I added further explanations to make the key messages unambiguous.

kind regards
